# Study of Xoconostle (*Opuntia* spp.) Powder as Source of Dietary Fiber and Antioxidants

**DOI:** 10.3390/foods9040403

**Published:** 2020-04-01

**Authors:** José Arias-Rico, Nelly del Socorro Cruz-Cansino, Montaña Cámara-Hurtado, Rebeca López-Froilán, María Luisa Pérez-Rodríguez, María de Cortes Sánchez-Mata, Osmar Antonio Jaramillo-Morales, Rosario Barrera-Gálvez, Esther Ramírez-Moreno

**Affiliations:** 1Área Académica de Enfermería, Instituto de Ciencias de la Salud, Universidad Autónoma del Estado de Hidalgo, 42000 Pachuca, Mexico; josearias.rico@hotmail.com (J.A.-R.); oajmorales@gmail.com (O.A.J.-M.); rosariobag@hotmail.com (R.B.-G.); 2Centro de Investigación Interdisciplinario, Área Académica de Nutrición, Instituto de Ciencias de la Salud, Universidad Autónoma del Estado de Hidalgo, 42000 Pachuca, Mexico; ncruz@uaeh.edu.mx; 3Departamento de Nutrición y Ciencia de los Alimentos, Facultad de Farmacia, Universidad Complutense de Madrid (UCM), 28040 Madrid, Spain; mcamara@farm.ucm.es (M.C.-H.); peromalu@ucm.es (M.L.P.-R.); cortesm@ucm.es (M.d.C.S.-M.)

**Keywords:** xoconostle, total dietary fiber, physicochemical properties, antioxidants

## Abstract

The objective of this study is to evaluate the nutritional composition, antioxidant properties, and functional characteristics of two cultivars of xoconostle *Opuntia xoconostle* F.A.C. Weber in Diguet cv. Cuaresmeño (XC) and *Opuntia matudae* Scheinvar cv. Rosa (XR). The samples were frozen (−32 °C, 48 h), lyophilized (96 h, −55 ± 1 °C, vacuum of 0.040 Mbar), and homogenized (size particle 500 μm) to get the xoconostle powder. Both cultivars (XC and XR) had a high content of carbohydrates characterized by soluble sugars (9.8 ± 0.7 and 29.9 ± 0.5 g/100 g dm) and dietary fiber (30.8 ± 0.7 and 36.8 ± 0.9 g/100 g dm), as well as lower proportions of organic acids, mainly citric acid (18.8 ± 0.0 and 13.6 ± 0.0 mg/100 g dm). These samples also had a high content of phenolic compounds (1580.3 ± 33.1 and 1068.5 ± 70.8 mg GAE/100 g dm), vitamin C (723.1 ± 16 and 320.2 ± 7.5 mg/100 g dm), and antioxidant activity ABTS·^+^ and DPPH· (between 1348.1 ± 74.0 and 3318.7 ± 178.8 µmol TE/100 g dm). Since xoconostle samples had a high content of dietary fiber, they were characterized by the capacity of water retention (water holding capacity 6.00 ± 0.1 and 5.5 ± 0.2 g H2O/g dm) and gel formation (swelling 5.2 ± 0.0 and 5.5 ± 0.0 g H2O/g dm), related with the retention of lipids and glucose in the food matrix similar to other foods. XR was characterized by a higher amount of dietary fiber, sugars and organic acids, while XC had higher phenols content and antioxidant properties, with higher values of functional properties. Then, our data suggest that both xoconostle cultivars in powder can be used as a functional ingredient for its fiber content and antioxidant properties, contributing with sensorial aspects as flavor and color. Therefore, these highly valued products can be used in the pharmaceutical and food industries.

## 1. Introduction

The genus *Opuntia* embraces about 1500 species of cactus [1,2], and it is largely distributed in America, Africa, and the Mediterranean. Species of the genus *Opuntia* generally produces tender young part of the cactus stem, known as cladodes, frequently consumed as a vegetable in salads, or fruits named cactus pear, characterized by an abundant sweet pulp. The fruits of the species, commonly known as xoconostles, stand out by its acidic taste [3]. Generally, cladodes are rich in pectin, mucilage, and minerals [4], while fruits are a good source of vitamins, aminoacids, organic acids, and betalains, contributing with attractive colors [5,6,7,8]. The great number of bioactive compounds, nutrients, and their functional properties make *Opuntia* spp. cladodes and cactus pear fruits, perfect candidates for the production of health-promoting food and supplements.

The xoconostle fruit grows from cactus paddles, and it is produced mainly in arid and semi-arid zones of Mexico, characterized by its hypocholesterolemic, hypoglycemic, and antihyperlipidemic properties [1,2]. Some species are barely exploited and are not marketed, as they are not very profitable for the farmers. In previous studies, different parts of xoconostle have been characterized: peel, seeds, and pulp have a high content of bioactive compounds [9,10,11,12,13]. However, only the mesocarp is considered as an edible portion, representing 70% of the fruit, while endocarp (seeds and pulp as mucilage) and peel are discarded [10]. The thin peel that corresponds to 25% of the total fruit is characterized by thorns of short length (13.7 ± 0.7 mm), white, arranged in a semi-erect position, and of flexible consistency [14], which are eliminated by a swept by the producer before the sale of the product. The consumption of this fruit in Mexico is limited, only it is used in sauces or in specific national dishes, due to its acidity, difficulties in postharvest handling and the lack of knowledge about its nutritional potential [15]. In addition, México and other Latinoamerican countries are undergoing a nutritional transition where indigenous foods (as xoconostle fruits) are underutilized. In this scenario, the powdered whole xoconostle could be employed as a natural source of dietary fiber, antioxidants, and food color, to be used as an ingredient in foods.

Therefore, the objective of the present study is the evaluation of two cultivars of whole xoconostle (including peel, seeds, and pulp) in powder as a potential source of dietary fiber and antioxidants, so that these fruits can be used as functional foods or as ingredients for nutritional supplements in the food or pharmaceutical industry.

## 2. Materials and Methods

### 2.1. Plant Materials and Sample Preparation

Two varieties of xoconostle were selected for this study: *Opuntia xoconostle* F.A.C. Weber in Diguet cv. Cuaresmeño (XC) and *Opuntia matudae* Scheinvar cv. Rosa (XR) (Figure 1). Samples were provided by the Mexican Association CoMeNTuna (Consejo Mexicano del Nopal y la Tuna) in Hidalgo, México. The xoconostle fruit harvested was grown in fields located at a latitude of 20°16′12″ N, longitude 98°56′42″ W, and altitude of 2600 m above sea level. The fruits were manually harvested during September 2015, in the optimum range of maturity for human consumption, considering the parameters of weight and diameter of the product, 20-30 days old, with a thin cuticle and the characteristic color of the variety, as established in Inglese Paole [16]. After harvesting, fruits were weighed and measured (weight, polar diameter, and equatorial diameter), and moisture was determined.

Then, the fruits were divided into three batches and were washed, sliced vertically (0.5 cm thickness) and frozen to −32 °C during 48 h and lyophilized at 96 h, −55 ± 1 °C with a vacuum of 0.040 Mbar, using a freeze dryer (VWR26671-581, Labconco). Samples were milled (Blender, 38BL52, LBC10, Waring Commercial) and sieved to a particle size of 500 μm. The powder was used for further analysis. The expression of results is as dry matter (dm).

### 2.2. Nutritional Composition

#### 2.2.1. Proximate Analysis

Analyses of moisture (Method 925.09), total protein (Method 950.48), fat (Method 983.23), and ash (Method 930.05) were carried out by the official methods described by the Association of Official Analytical Chemists [17].

#### 2.2.2. Total Soluble Sugars

Soluble sugars were determined by high performance liquid chromatography (HPLC), according to Sanchez-Mata et al. [18]. Five hundred mg of sample was extracted twice with 20 mL of 80% ethanol in a water bath maintained at a temperature of 55–60 °C for 45 min with constant stirring. After each extraction, the samples were centrifuged (Thermo Scientific, Sorval RC-5, Waltham) for 15 min at 1075× *g,* and the supernatants were pooled and filtered. The resultant extract was reduced in volume by using a rotary vacuum evaporator (R-200, Büchi) set to 70 °C to evaporate the ethanol. The concentrate was brought up to 5 mL with distilled water. Then, the samples were passed through a previously washed (5 mL of methanol followed by 5 mL of water) Sep-Pak C18 cartridge (Waters, Milford). Two milliliters of filtrate were mixed with 8 mL of acetonitrile to yield a total volume of 10 mL. Samples were filtered through a 0.45 μm Millipore membrane (Millipore, Bedford) before injection (200 μL aliquots) into the HPLC. The HPLC was equipped with a PU II isocratic pumping system (Micron Analitica, SA), a Rheodyne valve, and a differential refractometer R401 detector (Jasco). The chromatographic column used was a Luna 5l NH2 100 R (250 mm, 4.60 mm; Phenomenex, Torrance). The mobile phase was acetonitrile:water (80:20). Operating conditions were a flow rate of 0.9 mL/min and ambient temperature. All chromatograms were processed using Cromanec XP software (Micronec). The resultant peak areas in the chromatograms were plotted against concentrations obtained from standards of D(+)-sucrose, D(−)-fructose, and D(+)-glucose in a concentration range of 0.1–2 mg/mL.

#### 2.2.3. Total Dietary Fiber

Soluble (SDF) and insoluble (IDF) dietary fiber were determined according to AOAC [17], an enzymatic–gravimetric method using a Total Dietary Fiber Assay Kit (Sigma TDF-100A Kit, Sigma). Total dietary fiber (TDF) was the sum of SDF and IDF.

#### 2.2.4. Organic Acids and Vitamin C

The content of individual organic acids was quantified by high performance liquid chromatography (HPLC), based on Sánchez-Mata et al. [19]. Five hundred mg of each sample was homogenized with 25 mL of metaphosphoric acid (4.5%), with magnetic stirring for 15 min in the dark and centrifuged (9616× *g*, 15 min; Thermo Scientific, Sorval RC-5, Waltham). The supernatant was removed and pooled to 25 mL with metaphosphoric acid (4.5%), these extracts were filtered through filter paper and then through a PVDF filter of 0.45 μm for injection on HPLC.

An aliquot of extracts was subjected to reduction with L-cysteine to transform the dehydroascorbic acid (DHAA) in ascorbic acid (AA) and analyze the total vitamin C content. The instrument was a liquid chromatographer (Micron Analytical) equipped with an isocratic pump (PU II), an automatic injector (Jasco, AS-1555). The column used was a Sphereclone ODS 250 × 4.60, 5 μm (Phenomenex), and a UV–visible detector (Thermo Separation Spectra Series UV100). Chromatograms were processed using Cromanec XP software (Micronec). The conditions for the equipment were flow rate 0.4 mL/min at 215 nm and mobile phase of 1.8 mM H_2_SO_4_ (pH = 2.6) to determine organic acids. For the identification of ascorbic acid, the flow rate was 0.9 min/mL and 245 nm was set as the wavelength detection.

The organic acids found were quantified by comparison of the area of their peaks recorded at 215 nm with commercial standards (malic, citric, fumaric, and oxalic acid).

The AA was quantified by comparison of the area of their peaks recorded at 245 nm with calibration curves obtained from the commercial standard of AA. The oxidized form of vitamin C (DHAA) was calculated by the difference between total vitamin C and AA.

### 2.3. Antioxidant Properties

#### 2.3.1. Extraction of Antioxidants

The extraction of XC and XR were carried out by a modification of the method of Pérez-Jiménez et al. [20]. This method consists of the extraction of 250 mg of sample and extracted with 10 mL of MeOH/H_2_O (50:50, *v*/*v*), stirring during 30 min, and centrifugation (6678× *g*, 10 min; VanGuard V6500 Hamilton Bell^®^). Then, the supernatant was transferred to a flask of 25 mL, and the residue was re-extracted with 10 mL of acetone/H_2_O (70:30, *v*/*v*). Both supernatants were combined, and the flask was graduated to 50 mL with MeOH/H_2_O (50:50, *v*/*v*). The supernatants (methanolic and acetonic) were recovered, combined, and used to determine the total phenolic content and antioxidant capacity.

#### 2.3.2. Total Phenolic Compounds (TPC)

##### Folin–Ciocalteu

Total phenolic content was estimated based on the Folin–Ciocalteau procedure [21]. An aliquot of the extract solution (0.5 mL) was mixed with 0.5 mL of Folin–Ciocalteu reagent (2.5 mL, previously diluted with water 1:10 *v*/*v*) and sodium carbonate (75 g/L). After adding 0.4 mL of sodium carbonate, samples were allowed to stand for 30 min at room temperature, and the absorbance reading was performed at 765 nm. Gallic acid was used as a reference standard, and the results were expressed as milligrams of gallic acid equivalents per 100 g of dried matter (mg GAE/100 g dm).

##### Fast Blue BB (FBBB)

The procedure used was based on the modification reported by Maieves et al. [22]. Briefly, 4 mL of diluted samples (1:10) or standard were transferred to borosilicate tubes and were added with 0.4 mL of 0.1% FBBB reagent. After shaking, 0.4 mL of 5% NaOH were added, stirred, and let stand at room temperature for 90 min. Absorbance was measured at 420 nm in a Lambda 25 UV–vis spectrophotometer (Perkin Elmer). A mixture of 0.8 mL of deionized water plus 4 mL of sample in appropriated dilution was used as a blank of sample. The absorbance of samples was obtained by the difference between the absorbance of a sample and the absorbance of the sample blank. Gallic acid was used as a standard to build a calibration curve between 0 and 200 mg/L, in the same conditions as the samples. Total phenols were expressed as milligrams of gallic acid equivalent per 100 g of dried matter (mg GAE/100 g dm).

##### Quencher Fast Blue BB (Q-FBBB)

In order to optimize the TPC result in powdered fruits, a new methodology was employed using Fast Blue BB (FBBB) reagent, which reacts more specifically with phenolic compounds, along with direct QUENCHER [23]. The combination of both methodologies applied to solid matrices without extraction steps has been previously reported [24].

Briefly, 4.0 ± 0.5 mg sample were weighed in centrifuge tubes, with a subsequent addition of FBBB reagent (0.1% in distilled water, *m*/*v*) followed by addition of 0.4 mL of 5% NaOH (*m*/*v*), prepared with distilled water, and 4 mL of distilled water. After 1 h of incubation in an orbital shaker, the tubes were centrifuged at 5016× *g* for 25 min. Supernatants were filtered by gravity, and the absorbance was measured in each centrifuge tube at 420 nm in a Lambda 25 UV–vis spectrophotometer (Perkin Elmer). Furthermore, a control containing only the sample (4.0 ± 0.5 mg) with 4.8 mL of water was also prepared, with the aim of measuring the natural non-phenolic interfering compounds in the medium. Results were expressed as milligrams of gallic acid equivalents per 100 g of dry matter (mg GAE/100 g dm) by means of a dose-response curve for different quantities of the standard (0–155 mg/L).

#### 2.3.3. Antioxidant Capacity

##### ABTS·^+^ Methodology

Antiradical capacity was measured according to Kuskoski et al. [25]. Briefly, the radical cation (ABTS·^+^) was produced by reacting 7 mmol/L ABTS stock solution with 2.45 mmol/L potassium persulfate under dark conditions and room temperature for 16 h before use. The ABTS solution was diluted with deionized water to an absorbance of 0.70 ± 0.10 at 754 nm. Then, 20 µL of the extract of the sample was added to 980 µL of diluted ABTS solution; absorbance readings (754 nm) were taken in a microplate reader (Power Wave XS UV-Biotek, software KC Junior) after incubation for 7 min at room temperature. The antioxidant capacity was expressed as micromol of trolox equivalents per 100 g of dried matter (µmol TE/100 g dm).

##### DPPH· Methodology

The antiradical activity was measured with DPPH· radical as described by Morales and Jiménez-Pérez [26]. An ethanol solution (7.4 mg/100 mL) of the stable DPPH radical was prepared. Then, 100 µL of the sample was put into vials, and 500 µL of DPPH solution was added; the mixture was left to stand 1 h at room temperature. Finally, absorbance was measured at 520 nm using a microplate reader (Power Wave XS UV-Biotek, softwar KC Junior). Antiradical activity was expressed as micromol of trolox equivalents per 100 g of dry matter (µmol TE/100 g dm).

### 2.4. Physicochemical Characteristics

#### 2.4.1. Color Measurement

Color was measured using a Hunter Lab colorimeter (MiniScan XE™, Hunter Associates Laboratory Inc., Reston), using the D65 illuminant with an angle of observation of 10°. Color was recorded using the CIE–L∗a∗b∗ values, where L∗ indicates lightness (L∗ = 0 or 100 indicate black and white, respectively), a∗ is the axis of chromaticity between green (−) to red (+), and b∗ the axis between blue (−) to yellow (+). Numerical values of L∗, a∗ and b∗ were used to obtain hue angle (h°) (h° = tg^−1^ (b/a) and chroma (C = (a∗^2^ + b∗^2^)^1/2^) parameters [27].

#### 2.4.2. Hydration Properties

Water holding capacity (WHC), swelling (S), and oil holding capacity (OHC) were determined following the corresponding methodology for each determination according to Valencia and Román [28]

#### 2.4.3. Glucose Retention Index

To determine the glucose retention index (GRI), the modified methodology of Ou et al. [29] was applied. Samples (50 and 200 mg) were added to 10 mL of different glucose solutions (50, 100, and 200 mmol/L). The mixture was stirred and placed in a water bath with shaking at 37 °C for 6 h. After the incubation time, the samples were centrifuged (1454× *g*, 15 min) and the pellet was separated from the supernatant. Then, glucose content in the supernatant was determined by the DNS (3,5-dinitrosalicylic acid reagent) colorimetric method [30]. The difference in percentage between the concentration of glucose solution and the glucose concentration of the supernatant was considered as the glucose retention index.

### 2.5. Statistical Analysis

The results obtained were expressed as the mean ± standard deviation of at least three determinations. Data were analyzed using analysis of variance (ANOVA) with a significance level *p* < 0.05, *p* < 0.01, and *p* < 0.0001 to establish the differences between the two study samples. For this analysis, the statistical software Software GraphPad Prism version 5.03 was used.

## 3. Results and Discussion

The Cuaresmeño cultivar (XC) samples were characterized by a smaller size, weight (77.6 g), and polar diameter (5.8 cm), but with a larger equatorial diameter (4.9 cm) compared with Rosa cultivar (XR) (80.1 g, 7.0 cm, and 4.8 cm, respectively), with statistically significant differences in the values obtained for the polar and equatorial diameters of each variety (Figure 1). Parameters such as weight, polar and equatorial diameters of samples of both types of fruits were like those obtained for yellow xoconostle cultivar, evaluated by González-Ramos et al. [31], and higher than those obtained for xoconostles cv. Cuaresmeño from three different communities in Mexico, studied by Guzmán-Maldonado et al. [12].

### 3.1. Nutritional Composition

Table 1 shows the values obtained for the nutritional composition of the powdered whole xoconostle fruits (with peel and seeds) belonging to the two cultivars analyzed in this study. The dried samples were characterized by a high proportion of total carbohydrates constituted by soluble sugars and dietary fiber. In addition, the values obtained for XC were lower in comparison with XR, regarding soluble sugars (9.8 ± 0.7 and 29.9 ± 0.5 g/100 g dm) and fiber (30.8 ± 0.7 and 36.8 ± 0.9 g/100 g dm). These samples had a lower proportion of fat (2.5 ± 0.1 and 11.9 ± 0.1 g/100 g dm), proteins (4.0 ± 0.0 and 4.8 ± 0.1 g/100 g dm), and ashes (11.3 ± 2.9 and 11.7 ± 1.5 g/100 g dm). These values of xoconostle powder had the same tendency as the other study of the edible portion (peel and skin) of xoconostle fruit, where around 60% and 81% were constituted of total carbohydrates and 26.7% to 29.5% of total dietary fiber [12].

The powder sample of whole fruits studied had a lower content of soluble sugars than xoconostle endocarp reported in other studies [10,11]. It could be due to the fact that, in this study, samples included all the fruit, and the endocarp (characterized by a high amount of sugars) represents a low proportion of the fruit. On the other hand, the seeds which are constituted of total dietary fiber and starches (up to 70% starch) [32], and this component was not quantified in this study.

Glucose was the major soluble sugar in both samples with 6.8 ± 0.2 and 15.6 ± 0.2 g/100 g dm (XC and XR, respectively), followed by fructose and sucrose. The same tendency was reported in the endocarp in previous studies of the same cultivars [10,11]. Fructose was the major soluble sugar in pulp and seeds, and in the case of epicarp (peel); free sugars were found in lower amounts, with the predominance of sucrose [9].

The XC fruits had a lower content of TDF compared to XR, (30.8 ± 0.7 vs. 36.8 ± 0.9 g/100 g dm, *p* < 0.01), and both were characterized by a higher amount of insoluble, than soluble fiber. The results of TDF for XC and XR samples were higher than those reported by Guzmán-Maldonado et al. [12] in xoconostle (18.89, 10.63 and 0.95 g/100 g dm in peel, skin, and pulp, respectively). These differences in fiber content between studies could be due to the climatic conditions and soil characteristics during cultivation, as well as the stress conditions that occur during its storage.

On the other hand, due to the amount of fiber contained in the xoconostle cultivars (4 g/100 g fm in the fresh fruit or 30 g/100 g dm in the xoconostle powder), these samples can be considered as “source of dietary fiber” according to the classification established by Regulation of the European Parliament and Council (>3 g/100 g fm) [33]. The recommendations for consumption of TDF in adults, according to the American Dietetic Association (ADA), are 25–30 g/day, including a proportion of 3:1 for insoluble:soluble fiber, based on epidemiological studies that show protection against cardiovascular diseases [34]. According to this recommendation, 100 g of the fruit of both cultivars could cover up to 16% of the recommended intake of TDF per 100 g of fresh fruit, and comply with the suitable proportions for soluble and insoluble fiber. Therefore, the xoconostle powder samples could be used to increase the intake of dietary fiber as an ingredient in functional foods.

The XC powder presented a higher content of organic acids (20.8 ± 0.0 mg/100 g dm; malic, citric, fumaric, and oxalic), compared to the XR powder, that had 17.0 ± 0.0 mg/100 g dm (Table 1), similar to data reported by other studies [10,11]. High amounts of citric and malic acid contribute to the pH and flavor characteristic of these fruits.

With the methodology applied, it was possible to quantify the two active forms of vitamin C: ascorbic acid (AA) and dehydroascorbic acid (DHAA). According to the results obtained, a significant difference (*p* < 0.0001) was found in the total content of this vitamin, evidenced by a higher content of total vitamin C (86.8 mg/100 g fm or 723.1 ± 16.0 mg/100 g dm) in XC fruit powder, compared to the XR fruits (30.9 mg/100 g fm or 320.2 ± 7.5 mg/100 g dm).

According to the recommendations established by FAO [35] for daily consumption of this vitamin (75 to 80 mg), the two cultivars could be considered as “sources of vitamin C” (>12 mg/100 g fm of food), in accordance with the Regulation (EC) Number 1924/2006 [33], therefore, 100 g of these fruits could contribute with at least with 37% of established recommendations.

### 3.2. Antioxidant Properties

The content of total phenolic (Table 2) measured by Folin–Ciocalteu in XC and XR fruits (1580.3 ± 33.1 and 1068.5 ± 70.8 mg GAE/100 g dm, respectively) were similar to the results obtained by Guzman–Maldonado et al. [12], which is 128–986 mg GAE/100 g dm, distributed in the three different parts of the fruit (skin, pulp, and seeds) but concentrated in the fruit skin. In the same way, the values obtained of xoconostle powder were similar to the values found by Hernández-Fuentes et al. [3] (108 to 313 mg GAE/100 g fm, approximately 771–2235 mg GAE/100 g dm), in different xoconostle genotypes. Moreover, the phenolic content of xoconostle in this study was higher than those reported in dehydrated fruits such as apple, pear, papaya and mango (231.7–417.0 mg GAE/100 g dm), according with studies of Chong et al. [36]. It could be due to the fact that the peeling of the fruits prior to human consumption reduces the presence of antioxidant substances mainly located in the skin.

Fast Blue BB (FBBB) is a method developed by Medina [37], and it has been proposed for food labeling of total phenolic because there are no interferences of organic compounds naturally found in fruits and vegetables as it happens to the Folin–Ciocalteu method. However, the use of this routine method has been recently encouraged for its antioxidant potency in comparison with fruit-based beverages [38]. Total polyphenols measured by FBBB (XC 1415.2 ± 45.6 and XR 1088.5 ± 5.3 mg GAE/100 g dm) were comparable with the results obtained by Maieves et al. [22] in two first stages of maturation of *Hovenia dulcis* (1079 and 1778 mg GAE/100 g dm), and Medina [37] in dry blueberries (1291 mg GAE/100 g dm).

Among the different methodologies used, the highest values obtained for both samples corresponded to the Q-FBBB method. It should be noted that the Quencher protocol places in direct contact the solid material of the sample and the reagent solutions, allowing the phenols to bind to the insoluble matter by taking advantage of surface reactions occurring at the solid–liquid interface. This approach avoids time-consuming solvent extraction steps of the classical protocols [39] and allows the quantification of the so-called non-extractable polyphenols, mainly associated with insoluble fiber. Therefore, these values give a more reliable idea of the antioxidant effect expected after the ingestion of the whole product, since not only the extractable phenols can act as antioxidants in the gut. FBBB/Q-FBBB showed a high correlation (r = 0.98, *p* < 0.01) since the reaction between polyphenols and FBBB is the same, only varying the extraction methodology.

The ratio calculated between different methods of evaluation of total polyphenols FBBB/Folin–Ciocalteu, was 0.9 to XC and 1.0 to XR. The lower ratio in XC suggests the presence of higher levels of non-phenolic reducing constituents, for example, ascorbic acid, erroneously measured with the Folin–Ciocalteu method as “total phenolic” [40]. This fact is in agreement with a linear relationship found between Folin–Ciocalteu and ascorbic acid, established in the study of López-Froilán et al. [38].

Relative Antioxidant Capacity Index (RACI) has been proposed by Petrović et al. [41] and López-Froilan et al. [38] to achieve more comprehensive comparisons between analyzed samples, as well as applied assays. However, in this study, it was not necessary since only two varieties were compared.

The values obtained for the antioxidant activity determined by ABTS·^+^ (XC 3261.4 ± 102.7 and 1348.1 ± 74.0 µmol TE/100 g dm for XC and XR, respectively) and DPPH· (3318.7 ± 178.8 and 1753.5 ± 72.8 µmol TE/100 g dm for XC and XR, respectively) of the studied cultivars were higher and compared with other varieties of xoconostle [42], and similar to residues of cactus pear (around 3000 µmol TE/100 g dm for both methodologies) [43]. These differences in antioxidant activity between cultivars of xoconostle or cactus fruit may be related to the maturation of the fruit, which is associated with metabolic changes [44].

In addition, the fruits studied had similar values than those reported for dehydrated fruit such as apple, pear, papaya, mango, and blackberry (1244 to 3580 μmol TE/100 g dm) [36,45].

As can be seen, the XC powder presented a higher content of total phenols measured by different methodologies, and therefore antioxidant activity was higher (*p* < 0.0001). Therefore, a positive correlation of the antioxidant capacity with total phenols was found (r = 0.99, *p* <0.01), although other compounds play an important role in antioxidant activity, i.e., ascorbic acid.

### 3.3. Color

In general, the color of the fruits is one of the most important quality attributes that affect the final decision of purchase taken by the consumer. According to the location of the a* and b* coordinates, as well as hue angle of the studied samples, it is shown a yellow–orange tonality for the XC powder, and pink–red color for XR powder, being more intense and bright the color of XC, which indicates a higher value of L* and chromaticity (Table 3). Both samples had similar values of a*, b* and °h, lower brightness, and higher chromaticity between previous studies of the same cultivars [9,10,11]. Variations in color and luminosity depend on the content of certain compounds such as betacyanins (red–purple) and betaxanthins (yellow–orange), the state of maturity of the fruit at the time of harvest and post-harvest life, i.e., life during storage, in which fruit quality can be severely affected by water loss [44].

### 3.4. Functional Properties

Functional properties of dietary fiber or foods with a high content of fiber, such as the capacity of retention of different substances such as water, fat, and sugars within a food matrix, determine if it can be successfully incorporated in fiber-enriched foods. Fibers are considered food thickeners, stabilizers, and emulsifiers [46]. Not only dietary fiber helps evade hydrolysis, digestion, and absorption of oil and sugars in the human small intestine, but it achieves at least one of these functions: increases the faecal bulk, stimulates colonic fermentation, reduces postprandial blood glucose (reduces insulin responses), and reduces pre-prandial cholesterol levels [47,48].

This is the first report on the functional properties evaluated in xoconostle fruits powder. The powder samples analyzed presented significant differences between cultivars on these properties (Table 3). The powder products had water holding capacity (WHC) values of 6.0 ± 0.1 and 5.5 ± 0.2 g H_2_O/g dm, swelling (S) of 5.2 ± 0.0 and 5.5 ± 0.0 g H_2_O/g dm, and an oil holding capacity (OHC) of 6.1 ± 0.3 and 3.4 ± 0.2 g of oil/g dm for XC and XR, respectively. The values obtained for WHC in the cultivars of the present study were similar to the artichoke flour evaluated by López et al. [49] and apple pomace flour [50] but higher than those obtained for grapefruit, lemon, orange, and apple pomace (≈1.6 to 2.3 g H_2_O/g dm) [51]. Swelling property (S) in the studied cultivars were similar to those reported by Figuerola et al. [51] in orange pomace as well as apple pomace flour [50], with values ≈6 g H_2_O/g dm, while OHC of both cultivars were higher in comparison with the study previously mentioned [50,51].

A series of different concentrations of glucose was used to evaluate the glucose retention capacity of the sample characterized by a high content of fiber. The results showed that the samples of powdered xoconostle could bind glucose in solutions of different concentrations, and, therefore, it could be suggested that available glucose in the small intestine could be kept at a lower concentration by the presence of dietary fibers [29]. This behavior had been related to the hypoglycemic properties of the xoconostle fruit, according to Pimienta-Barrios et al. [1], who analyzed the effect of the ingestion of the peel of xoconostle in healthy people and in people with type 2 diabetes, resulting in a decrease in serum glucose concentration, as well as an increase in insulin. The values of GRI were lower compared with different dietary fibers [29] and those obtained by Abirami et al. [52] for the fiber extracts of peel, pulp, and seeds of the lime. These differences could be due to the fact that these authors studied the fiber extracts, while, in this study, the whole fruit was analyzed. This is an advantage because when using the whole fruit, it provides not only fiber dietary and its properties, but also those of other components such as antioxidants, organic acids, color compounds, among others. In addition, the response of the functional properties of the xoconostle powder could be affected by the particle size that we use (500 μm). According to Zlatanović et al. [53], the functional properties as swelling, oil holding, and hydration capacity are reduced with the decrease of particle size. Raghavendra et al. [54] established that the range to preserve these functional properties is between 550 to 1127 μm.

Finally, given the high nutritional potential of freeze-dried xoconostle fruits, as it has been explored for other food ingredients [50], further research should also be done to evaluate shelf-life, nutritional potential, and functional properties of the desiccated xoconostle product obtained using other techniques, such as low temperature desiccation.

## 4. Conclusions

The whole xoconostle fruits (cv. Cuaresmeño and Rosa), as a powder, without separate peel, and seeds (these parts considered as waste), can be used as a functional ingredient that provides dietary fiber, antioxidants, soluble sugars, and organic acids with great importance in the pharmaceutical and food industry.

## Figures and Tables

**Figure 1 foods-09-00403-f001:**
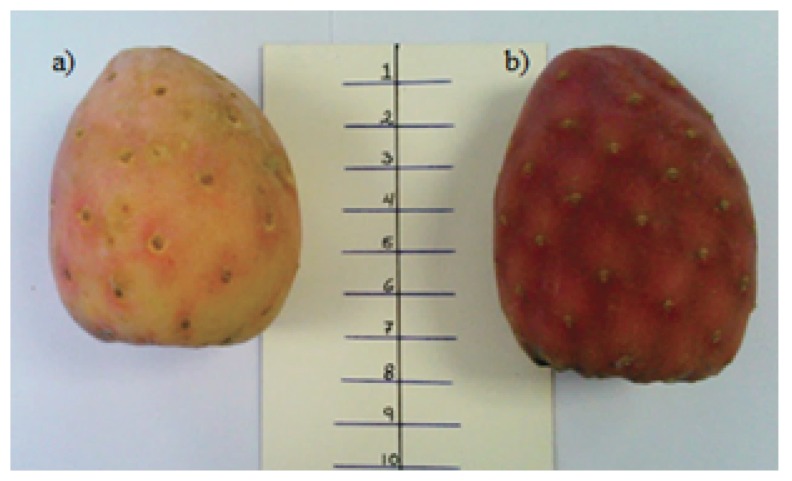
Images of xoconostle fruits: (**a**) *Opuntia xoconostle* F.A.C. Weber in Diguet cv. Cuaresmeño, (**b**) *Opuntia matudae Scheinvar* cv. Rosa (scale is in centimeters).

**Table 1 foods-09-00403-t001:** Nutritional composition of two cultivars of xoconostle ^A^.

	cv Cuaresmeño	cv Rosa
Moisture of fresh fruit	90.4 ± 0.7 **	85.9 ± 1.3
Moisture	7.2 ± 0.1 **	8.2 ± 0.2
Protein	4.0 ± 0.0 ***	4.8 ± 0.1
Fat	2.5 ± 0.1 ***	11.9 ± 0.1
Ash	11.3 ± 2.9	11.7 ± 1.5
Total soluble sugars	9.8 ± 0.7 ***	29.9 ± 0.5
Sucrose	1.5 ± 0.2 ***	6.3 ± 0.1
Fructose	1.5 ± 0.4 ***	8.0 ± 0.3
Glucose	6.8 ± 0.2 **	15.6 ± 0.2
Total dietary fiber	30.8 ± 0.7 ***	36.8 ± 0.9
Soluble fiber	8.2 ± 0.4 **	5.8 ± 0.4
Insoluble fiber	22.6 ± 0.4 ***	31.0 ± 0.6
Organic acids ^B^	20.8 ± 0.0 ***	17.0 ± 0.0
Malic acid	1.3 ± 0.0 ***	2.9 ± 0.0
Citric acid	18.9 ± 0.0 ***	13.6 ± 0.0
Fumaric acid	0.1 ± 0.0 ***	0.1 ± 0.0
Oxalic acid	0.6 ± 0.0 ***	0.5 ± 0.0
Total vitamin C ^B^	723.1 ± 16.0 ***	320.2 ± 7.5
Ascorbic acid (AA)	169.4 ± 2.7 ***	385.5 ± 6.1
Dehydroascorbic acid (DHAA)	337.6 ± 10.8 ***	150.8 ± 6.1

^A^ Results (mean ± standard deviation, *n* = 3) are expressed as g/100 g dm except moisture of fresh fruit. ^B^ Results are expressed as mg/100 g dm. The asterisk in the column reports statistically significant differences between the varieties (** *p* < 0.01, *** *p* < 0.0001).

**Table 2 foods-09-00403-t002:** Antioxidant properties of two cultivars of xoconostle.

	cv Cuaresmeño	cv Rosa
Total phenolic compounds ^A^		
Folin–Ciocalteu	1580.3 ± 33.1 ***	1068.5 ± 70.8
FBBB	1415.2 ± 45.6 ***	1088.5 ± 5.3
Q-FBBB	1752.4 ± 21.5 **	1438.5 ± 71.9
Ascorbic acid ^B^	169.4 ± 2.7 ***	385.5 ± 6.1
ABTS·^+^ ^C^	3261.4 ± 102.7 ***	1348.1 ± 74.0
DPPH·^C^	3318.7 ± 178.8 ***	1753.5 ± 72.8

^A^ The results are expressed in mg GAE/100 g dm. ^B^ The results are expressed in mg AAE/100 g dm. ^C^ The results are expressed in µmol TE/100 g dm. The asterisk in the column reports statistically significant differences between the varieties (** *p* < 0.01, *** *p* < 0.0001).

**Table 3 foods-09-00403-t003:** Physicochemical characteristics of two cultivars of xoconostle.

	cv Cuaresmeño	cv Rosa
Color		
Lightness (*L**)	33.5 ± 0.1 ***	22.4 ± 0.0
Intensity of red (*a**)	5.3 ± 0.0 ***	10.4 ± 0.0
Intensity of yellow (*b**)	14.0 ± 0.0 ***	9.6 ± 0.0
*° h*	69.2 ± 0.1 ***	42.7 ± 0.0
*Chroma*	15.0 ± 0.0 ***	14.2 ± 0.0
Water holding capacity (WHC) ^A^	6.0 ± 0.1 *	5.5 ± 0.2
Swelling (S) ^A^	5.2 ± 0.0	5.5 ± 0.0
Oil holding capacity (OHC) ^B^	6.1 ± 0.3 *	3.4 ± 0.2
Glucose retention index (GRI) ^C^		
Glucose concentration		
50 mmol/L	48.6 ± 0.1 *	48.1 ± 0.0
100 mmol/L	96.8 ± 0.1 ***	95.1 ± 0.2
200 mmol/L	189.2 ± 0.5	189.8 ± 0.0

Brightness 0–100, a* (chromaticity axis from green (−) to red (+)), b* (chromaticity axis from blue (−) to yellow (+)), ° h (hue angle (tonality)) and chroma (Chromaticity (color saturation)). ^A^ Results are expressed as g H_2_O/g dm. ^B^ Results are expressed as g oil/g dm. ^C^ Results are expressed as µmol glucose/g dm. The asterisk in the column reports statistically significant differences between the varieties (* *p* < 0.05, *** *p* < 0.0001)

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
