# Peer review of "Study of Xoconostle (*Opuntia* spp.) Powder as Source of Dietary Fiber and Antioxidants"

_foods, 2020, doi:10.3390/foods9040403_

Round 1
Reviewer 1 Report
This manuscript reports the partial physico-chemical characterization of the fruit of a Mexican cactus. Sound methods were followed and the data are presented in a reasonable way. In this reviewer’s opinion, some revision may improve the clarity and usefulness of the report. The authors are invited to consider the following points:
-The article deals with a species of Opuntia genus which seems barely available outside Mexico. This contrasts with the Opuntia species known as “nopal” which are gaining wider attention, particularly in Southern Europe. How different may Xoconostle and nopal be? Could Xoconostle be a better “functional” alternative to nopal from an international perspective? With this in mind, it would be beneficial to include comments comparing the present data with those known for nopal. In this context, the recent comprehensive review of nutritional and functional features of Opuntia cactuses published by Dr Paredes-Lopez’ group is worth citing: Nopal (Opuntia spp.) and its Effects on Metabolic Syndrome: New Insights for the Use of a Millenary Plant. Angulo-Bejarano et al. Curr Pharm Des. 2019;25(32):3457-3477. doi: 10.2174/1381612825666191010171819 ).
-Abstract, lines 35-38: since the article reports compositional data only, it does not seem prudent to “suggest that …xoconostle should be promoted as a healthier diet”. Moreover, “attractive colors” cannot be the basis for including a plant ingredient in nutritional supplements. Please modulate the outro conclusions accordingly.
-Lines 75-85: how many batches of each fruit were analyzed separately? Or was it only a pool of fruits supplied by CoMeNTuna? This should be clarified and commented in the Discussion regarding the generalizability of the data being reported.
-Lines 328-332: please revise this paragraph since it does not seem to convey a clear message. What was meant by “chewing to stool formation related WHC”? and what is “reabsorption of nutrients” ?
-Lines 332-333: given the analytical nature of the study, the conclusion stating that “the fiber contained in these fruits could be recommended in the treatment of overweight, obesity and diseases related to the gastrointestinal tract” does not seem to have enough support and should be reformulated. This is also applicable to lines 410-413 in the Conclusions section.
-Tables 1 and 2: the +/- signs between average and SD values are missing.
-Table 2: The values for “Total soluble sugars” are missing. In addition, the footnote states that results are expressed as mg/100 g fw; is that correct? It does not seem right for the proximate constituents and dietary fiber contents.
Author Response
We are pleased to acknowledge receipt of the suggestions of the reviewers. According to your instructions, we revised the paper and took into account the changes and suggestions of the reviewers in our manuscript. The manuscript has been altered according to the comments made by the reviewers. You can find all changes and responses to each comment made by the reviewers in the following lines. Also the Checklist for style was followed carefully. The manuscript has doubtless been very much improved after revision according to the reviewers’ suggestions.
We would like to thank you and the reviewers for your very helpful comments.

Reviewer 2 Report
General comments
The submitted ms should be changed to offer more relevant insight in functionality of powder obtained, additional results needs to be included and discussed, multiple mistakes have to be corrected, some explanations provided etc.
Title
The title does not correspond with the content of the submitted ms. It should be clear that fruit, dehydrated by freeze drying is analyzed.
For example Nutritional and functional properties of the powder obtained by freeze drying of two varieties of … would be more suitable
Abstract
Abstract needs to be more informative and precise. Method of drying needs to be mentioned, it should be stressed that powder obtained by freeze drying was analysed, results need to be expressed per dry weight, dried fruit is extracted and analysed, not a fresh one. It needs to be clear that nutritional and functional properties of the powder are considered. After proximate composition and DF content, AO activity determined by the application of multiple methods should be mentioned, followed by a colour and functional properties et the end. Determined values obtained for WHC and OHC could be given. They depend mostly on DF but WHC and OHC of whole powder of the fruit was determined. Thus, authors should avoid term physicochemical properties of fiber. Functional properties of the powder obtained by freeze drying woudl be correct and more appropriate.
First two sentences should be avoided. The last one needs to rewriten or replaced since it brings confusion. Healtier in comparission with what? Suggestion to introduce colour as a nutritional supplement is rather confusing. The whole text of abstract should be rewritten. Sentences related to influence on public health should be avoided. However, obtained powder potential to be used as dietary supplement or ingredient of functional food could be mentioned in abstract and discussed in RD.
Introduction
Introduction and the whole paper should be changed in accordance with the fact that powder was the subject of the study conducted.
Experimental
Experimental is not precise and clear enough, some parts are rather confusing, important details are missing. For the sake of clarity missing details needs to be provided and range of mistakes corrected. The rank of analysis should be as follows nutritional properties, i.e.proximate composition, DF, TPC, AO activity, colour which is often in correlation with TPC, and finally functional properties such as WHC, OHC, GRI. All results need to be expressed per dry weight.
2.1. Full Information regarding the sampling of fruit needs to be given. If fruit from only one location was collected at one time point? How the sampling was performed? How the maturity of the fruit was estimated?
Why the fruit was not cut into smaller pieces to enable faster drying? Duration of lyophilisation needs to be expressed in hours, not in days. It is not clear how the dehydrated fruit was grinded. Homogenization and homogeneous is not correct term and should be avoided from 2.1. Why obtained flour was sieved?
2.2.2. Please describe how the powder was extracted and which quantity of liquid is used. Instead of volume of aliquot of extract, weight of powder is given!!!
Abbreviations used in MM are not the same as used in Tables, for example DHAA in MM DHA in Table, S in MM CH in Table etc., and some are not properly introduced,
2.3.2. It should be mentioned how results are expressed
2.4.1. Please explain why methanol and acetone are chosen as solvents. It is preferable to use food and ecological compatible solvents.
Instead of physicochemical term functional properties could be used. This part can be extended. Particle distribution and additonal functional properties can be determined in order to give a better insight in possibility of powder application, such as in paper recently published in Foods (2019, 8,561 https://doi.org/10.3390/foods8110561).
2.7. Since two AO assays were used and some TPC assays used are not specific for phenolic but known to determine total reducing power, calculation of Relative Antioxidant Capacity Index (RACI) can be considered (Combinatorial Chemistry and High Throughput Methods, 2016;19 (1): 58-65). However, it is more important to be aware that instead of multiple assays for TPC it would be more usefull to identify individual phenolic compounds present. To provide a better insight in functionality of powder authors should rather provide results related to identification and quantification of phenolics.
Results
Table 1 can be avoided.
3.1. Table 2 Please provide explanation for so high moisture content. After 4 days of drying it is rather high. Please provide explanation for such big difference, 85 and 90%, as well. Activity of water as a very important parameter of stability could be determined and included.
3.3. Expression such as These products had a WHC between should be avoided having in mind low number of samples. There are only two products, i.e. only two samples, there is nothing between values determined for these two samples.
Table 3 Please be consistent with abbreviations used. They differ in MM and Tables.
A number of significant numbers should be corrected to be always the same, not 9.59 ±0. 003 but 9.6±0.1, not 95.1±0.17 but 95.1±0.2 etc.
Delete Glucose retardation index above Water holding capacity
Abbreviation for Swelling is S or CH?
The glucose retention index is expressed in mmol/L in the Table 3, in line 310 µmol/L stands while in line 317 92.38 i 90.74%. !!!!
315 316 Higher than what? !
For comparative reasons functional properties of apple pomace flour (Foods 2019, 8,56) as rich source of DF can be introduced in discussion
243 Delete ...which characteristics of most food of vegetable origin.
Since the powder of whole fruit has much longer shelf life than fresh fruit preservation of the fruit by desiccation at low temperatures is a good idea and can have potential. Possibility of dried fruit or powder production and placing on the market could be considered. Since freeze drying is time consuming and costly method, not widely available especally at industrial scale level, other methods able to preserve nutritional and functional value could be considered in authors further work. Investigation of the stability of the obtained powder or flour can be the subject of further work as well. It is important to get an insight in shelf life of powder, thermal stability (temperature of glass transition) (see Thermochimica Acta, 673, 2019,17-25), water activity, phenolics retention etc
Author Response
Comments and Suggestions for Authors
General comments
The submitted ms should be changed to offer more relevant insight in functionality of powder obtained, additional results needs to be included and discussed, multiple mistakes have to be corrected, some explanations provided etc.
Title
The title does not correspond with the content of the submitted ms. It should be clear that fruit, dehydrated by freeze drying is analyzed.
For example Nutritional and functional properties of the powder obtained by freeze drying of two varieties of … would be more suitable
Answer: We have changed the title of paper according the recommendation of reviewer considering the dates in dry material (DM) as an functional ingredient of foods with the follow title:
“Study of Xoconostle (Opuntia, spp) fruit powder as source of dietary fiber and antioxidants”
Abstract
Abstract needs to be more informative and precise. Method of drying needs to be mentioned, it should be stressed that powder obtained by freeze drying was analysed, results need to be expressed per dry weight, dried fruit is extracted and analysed, not a fresh one.
Freeze-drying, or lyophilization, is a methodology used in different studies to preserve samples during the storage and analysis period, and it is not used as a extraction method of bioactive compounds. In the other hand, we can calculate the nutritional and antioxidant composition in fresh or dry material, in order that we can compare with other studies. Date in fresh material is important due to composition of food is as they are eaten. However we are agree with the recommendations of the reviewer and we have considered expressed the dates in dry material to highlight the xoconostle in powder as an ingredient in the pharmaceutical and food industry.
Answer: According the recommendation of reviewer, the methods of freeze drying has been described in the abstract line 28-32 and in the methodology line 89-93
It needs to be clear that nutritional and functional properties of the powder are considered.
Answer: This recommendation has included in abstract in line 41-44 as follow::
“Our data suggest that the xoconostle powder could be used as a functional ingredient providing dietary fiber, sugars, organic acids and antioxidant compounds that contribute with sensorial aspects as flavor and color with great importance in pharmaceutical and food industry”.
After proximate composition and DF content, AO activity determined by the application of multiple methods should be mentioned, followed by a colour and functional properties et the end.
Answer: We reorganized the methods and results considering first the composition and the final the functional properties, according the recommendation of reviewer.
Determined values obtained for WHC and OHC could be given. They depend mostly on DF but WHC and OHC of whole powder of the fruit was determined. Thus, authors should avoid term physicochemical properties of fiber. Functional properties of the powder obtained by freeze drying woudl be correct and more appropriate.
Answer: We agree with the reviewer that the functional properties of xoconostle fruit are result of the product in powder and not specifically the fiber. Therefore, the paragraph in section 3.5 has been corrected.
First two sentences should be avoided. The last one needs to rewriten or replaced since it brings confusion. Healtier in comparission with what? Suggestion to introduce colour as a nutritional supplement is rather confusing. The whole text of abstract should be rewritten. Sentences related to influence on public health should be avoided. However, obtained powder potential to be used as dietary supplement or ingredient of functional food could be mentioned in abstract and discussed in RD.
Answer: The whole text of abstract had been rewritten, avoiding the influence on public health and remarking the part of advantages of a fruit powder as a functional ingredient.
Introduction
Introduction and the whole paper should be changed in accordance with the fact that powder was the subject of the study conducted.
Answer: According the suggestion of reviewer, we have changed the paper considering the fruit powder as subject the study.
Experimental
Experimental is not precise and clear enough, some parts are rather confusing, important details are missing. For the sake of clarity missing details needs to be provided and range of mistakes corrected. The rank of analysis should be as follows nutritional properties, i.e.proximate composition, DF, TPC, AO activity, colour which is often in correlation with TPC, and finally functional properties such as WHC, OHC, GRI. All results need to be expressed per dry weight.
Answer: We reorganized the methods and results considering first the composition and the final the functional properties, and all the results are shown as dry matter according the recommendation of reviewer.
2.1. Full Information regarding the sampling of fruit needs to be given. If fruit from only one location was collected at one time point? How the sampling was performed? How the maturity of the fruit was estimated?
Answer: All the information asked by the reviewer has been included in the section 2.1. Plant material and sample preparation.
Why the fruit was not cut into smaller pieces to enable faster drying? Duration of lyophilisation needs to be expressed in hours, not in days. It is not clear how the dehydrated fruit was grinded. Homogenization and homogeneous is not correct term and should be avoided from 2.1. Why obtained flour was sieved?
Answer: All the information has been included in the section 2.1. Line 79-93. Effectively, the fruit was cut in small slices to enable faster drying. In the methodology we changed the time of day in hours. The grinded was performed with a Blender, 38BL52, LBC10, Waring Commercial, USA. Respect to sieved at particle size of 500 μm, the importance of this step is having a homogeneous material. In addition, several authors (Guillon & Champ, 2000) have established that the properties that are nutritionally relevant are affected mainly the particle size between others (bulk volume, the surface area characteristics, the hydration and rheological properties and the adsorption or entrapment of minerals and organic molecules), therefore the importance of know the size particle of the study sample.
Guillon, F., & Champ, M. (2000). Structural and physical properties of dietary fibres, and consequences of processing on human physiology. Food research international, 33(3-4), 233-245.
2.2.2. Please describe how the powder was extracted and which quantity of liquid is used. Instead of volume of aliquot of extract, weight of powder is given!!!
Answer: We had described how the powder was extracted to determine soluble sugars in the section 2.2.2. Line 100-118
Abbreviations used in MM are not the same as used in Tables, for example DHAA in MM DHA in Table, S in MM CH in Table etc., and some are not properly introduced,
Answer: The paper was completely revised and corrected
2.3.2. It should be mentioned how results are expressed
Answer: The information has been included in the section 2.1.
2.4.1. Please explain why methanol and acetone are chosen as solvents. It is preferable to use food and ecological compatible solvents.
Answer: We have publishes results about different food matrix from extraction methodology by methanol/water followed by acetone/water having good results.
This methodology was validated by other author, which indicate that ”A procedure for extraction of antioxidants from plant foods should combine at least two extraction cycles performed with aqueous-organic solvents with different polarities in order to extract antioxidant compounds with different chemical structures. A general procedure is routinely used at our lab to extract antioxidants from different foodstuffs, including extraction with acidic methanol/water (50:50, v/v; pH 2), followed by acetone/water (70:30, v/v)”
- Larrauri, J. A., Rupérez, P., & Saura-Calixto, F. (1997). Effect of drying temperature on the stability of polyphenols and antioxidant activity of red grape pomace peels. Journal of Agricultural and Food Chemistry, 45(4), 1390–1393.
- Pérez-Jiménez, J., & Saura-Calixto, F. (2006). Effect of solvent and certain food constituents on different antioxidant capacity assays. Food Research International, 39, 791–800.
- Saura-Calixto, F., & Goñi, I. (2006). Antioxidant capacity of the Spanish Mediterranean diet. Food Chemistry, 94(3), 442–447
However we agree that improvements should be done in this area to use better ecologically compatible solvents for food analysis.
Instead of physicochemical term functional properties could be used. This part can be extended. Particle distribution and additional functional properties can be determined in order to give a better insight in possibility of powder application, such as in paper recently published in Foods (2019, 8,561 https://doi.org/10.3390/foods8110561).
Answer: The recommended paper by the reviewer was included and the influence of the size particle in the functional properties was described more widely as follow in the lines 421-425
the response of the functional properties of the xoconostle powder could be affected by the size particle that we use (500 μm). According to Zlatanović, et al [55], the functional properties as swelling, oil holding, and hydration capacity are reduced with the decrease of particle size. Even, Raghavendra et al [56] established that the range to preserve these functional properties is between 1127 to 550 μm.
2.7. Since two AO assays were used and some TPC assays used are not specific for phenolic but known to determine total reducing power, calculation of Relative Antioxidant Capacity Index (RACI) can be considered (Combinatorial Chemistry and High Throughput Methods, 2016;19 (1): 58-65).
Answer: An explanation about the interest and possibility of calculating the RACI has been included in the manuscript (see page 9 line 350-354) and a revised list of references has been submitted. In this study we have not intended to calculate the RACI because the number of samples to be compared is only two. We have calculated a similar index in a previous study (Food Analytical Methods (2018) 11:2897–2906 https://doi.org/10.1007/s12161-018-1259-1).
However, it is more important to be aware that instead of multiple assays for TPC it would be more useful to identify individual phenolic compounds present. To provide a better insight in functionality of powder authors should rather provide results related to identification and quantification of phenolics.
Answer: Our objective in this case was not to study the individual phenolic compounds of samples but we agree with the reviewer's opinion about of the interest of individual compounds analysis. We will be considered that it could be performed in future studies.
Results
Table 1 can be avoided.
Answer: The table 1 have been eliminated and the dates have been described in the results only.
3.1. Table 2 Please provide explanation for so high moisture content. After 4 days of drying it is rather high. Please provide explanation for such big difference, 85 and 90%, as well. Activity of water as a very important parameter of stability could be determined and included.
Answer: The dates of the table 1 have been calculated in fresh weight, therefore, the moisture is high (85-90%). The moisture is calculated with the fresh sample according with the methodology of the Association of Official Analytical Chemist. The other dates are obtained in dry weight, and with the date of moisture of the food we can calculate the percentage of the component in fresh weight. However, we had changed the dates to dry material where the moisture is around of 7%. We did not do the water activity. We agree that is a very important parameter of stability and will be considered in further studies.
3.3. Expression such as These products had a WHC between should be avoided having in mind low number of samples. There are only two products, i.e. only two samples, there is nothing between values determined for these two samples.
Answer:The sentences that express ranges of values had been avoid, in order that no generalize values of the xoconostle.
Table 3 Please be consistent with abbreviations used. They differ in MM and Tables.
Answer: The abbreviations had been corrected in all the document.
A number of significant numbers should be corrected to be always the same, not 9.59 ±0. 003 but 9.6±0.1, not 95.1±0.17 but 95.1±0.2 etc.
Answer: The number of significant numbers had been corrected and homogenized to two digits after of point in all the document.
Delete Glucose retardation index above Water holding capacity
Answer: We had eliminated the term in the table 2.
Abbreviation for Swelling is S or CH?
Answer: The abbreviations of swelling is S. We had been corrected in the entire document.
The glucose retention index is expressed in mmol/L in the Table 3, in line 310 µmol/L stands while in line 317 92.38 i 90.74%. !!!!
Answer: The units had been corrected in the table 3 and in the description of the results. The units of glucose concentration are in mmol while the results are in µmol glucose .
315 316 Higher than what? !
Answer: The word higher was eliminated of the description in the line 400, due in the next paragraph it is compared with other samples.
For comparative reasons functional properties of apple pomace flour (Foods 2019, 8,56) as rich source of DF can be introduced in discussion
Answer: The recommended paper by the reviewer was included and the values of functional properties was compared with our study in the lines :405-408, 422-425
243 Delete ...which characteristics of most food of vegetable origin.
Answer: The sentence had been eliminated.
Since the powder of whole fruit has much longer shelf life than fresh fruit preservation of the fruit by desiccation at low temperatures is a good idea and can have potential. Possibility of dried fruit or powder production and placing on the market could be considered. Since freeze drying is time consuming and costly method, not widely available especially at industrial scale level, other methods able to preserve nutritional and functional value could be considered in authors further work. Investigation of the stability of the obtained powder or flour can be the subject of further work as well. It is important to get an insight in shelf life of powder, thermal stability (temperature of glass transition) (see Thermochimica Acta, 673, 2019,17-25), water activity, phenolics retention etc
Answer: We thank the reviewer for this comment. Freeze-drying is the best way to preserve food products, since it use very low temperatures and vacuum conditions, and thus nutrients are best preserved than using other different techniques; for that reason it has been used in this work, mainly to preserve the samples through the time to the analysis lasted. But also, it is a way to obtain stabilized ingredients for food industry, so the parameters measured are useful information to know what is inside the freeze-dried ingredient. Of course, low temperature desiccation may be also suitable, as it is a less expensive technique. The study of stability of nutrients in dessiccated xoconostle fruit using different methodologies could be a very interesting research to be conducted, based on the reference given. We will explore about that for a near future research.
Round 2
Reviewer 2 Report
General comments for authors
The ms is improved but still does not have satisfied quality. Several suggestion can be given to provide novelty, for example to include investigations related to stability of powder during the storage, i.e. measurement of TPC; AO activity and color upon or during storage, as well as surveying of water activity changes.
The style of writing is inappropriate. Comments on Abstract are only an example that should give idea how to improve the whole text to be concise and adequate for publishing.
Abstract
Abstract of 350 words needs to be shortened for the sake of conciseness, exactness and clarity. It needs to be completely rewritten and checked by native speaker with experience in scientific writting.
Following sentences “Mexico is undergoing an epidemiological and nutritional transition that generates an increase in the national morbidity and mortality rate. This effect can be reduced by the promotion of a varied and inclusive diet including indigenous foods, such as underutilized xoconostle fruit, with a content of bioactive compounds, which contributes to prevention of different diseases.” (50 words) needs to be avoided since relation between national mortality and an attempt to dehydrate local fruit by time consuming and very costly freeze drying is not remarkable enough.
There are many unnecessary parts in Abstract. Following information such as that fruits were provided by an association of farmers, that all the data were expressed as dry matter etc are uncommon to exist in Abstract and should be avoided as well.
The sentence “Xoconostle fruits were characterized by an attractive color between yellow-orange (cv Cuaresmeño) and reddish rose (cv. 33 Rosa)” does not provide info about the work done and should not exist.
Also widely accepted facts needs to be avoided such as “it is more important to take advantage of the component in the whole dehydrated product than when the extract of the fruits is used”. Also, this is not a topic and any such comparison was made during the study conducted.
The sentence “The powder fruits had properties for the formation of gels, retention of water, lipids and glucose” should be rewritten since fruits powder generally have such properties. Specific data obtained for analyzed fruits powders needs to be given instead.
The following sentence “Therefore, the samples could be considered sources of fiber and vitamin C” should be deleted since the last sentence in Abstract contains its repetition.
Also, the sentence “These samples also had a high content of phenolic compounds, which together with ascorbic acid, contributing to their antioxidant activity” should be rewritten. It is widely known that phenolics and ascorbic acid contribute AO activity, authors did not consider specific contribution of individual compounds to total AO activity in their work…..
The last sentence could be shorter. Expression such as great importance needs to be avoided.
More exact data should be introduced, for example duration of drying and temperature can be included in part of abstract related to drying (in brackets), dry matter of powder obtained (also in bracket), not with details, just number (For example Moisture of ….was achieved upon 96 h of freeze drying at temperature of ... ), data related to ratio of soluble and insoluble fibers, exact values of WHC i OHC, as well as FC, ABTS and DPPH. Some differences between cultivars might be mentioned, if significant enough.
MM
There are many confused parts of MM, with missing details or parts different than original version
For example, In original version vacuum was 0.040, in revised version 0.140?
Quantity (volume) of acid is missing in Five hundred mg of each sample were homogenized with metaphosphoric acid (4.5%), with magnetic stirring for 15 min in the dark and 126 centrifuged (9616 g, 15 min) (Thermo Scientific, Sorval RC-5, Waltham).
Since it would be extremely time consuming for reviewers to introduce all corrections necessary only some examples are given, there are numerous incorrectness.
RD
For the sake of clarity many parts of RD should be changed as well.
In Table 1 the values for ash are impossible for the stats to be significantly different. Please check the stats again!
Also, not all values should have 2 decimal places, but the last significant digit, it is impossible for all methods to have the same last significant digit, this applies to all tables.
244-250, it is not clear what values correspond to which sample
The last sentence in this paragraph is completely unclear
Expressing results such as ≈ 30 g/100g dm is not acceptable. Exact numbers should be given, with SD included.
Line 275 unit is missing after 100
Extensive checking of units through text is strongly recommended.
Conclusion
Conclusion should be rewritten, the last sentence deleted.
Author Response
March 13th, 2020
Editor in Chief
Foods Editorial Office
Manuscript No: foods-725094
Dear Editor,
Please find enclosed the revised version of the manuscript entitled “Study of Xoconostle (Opuntia, spp) Powder as Source of Dietary Fiber and Antioxidant” by José Arias-Rico, Nelly Cruz-Cansino, Montaña Cámara-Hurtado, Rebeca López-Froilán, María Luisa Pérez-Rodríguez, María de Cortes Sánchez-Mata, Osmar Antonio Jaramillo-Morales, Rosario Barrera-Gálvez and Esther Ramírez-Moreno, that we would like to be reconsidered for publication in the Journal Foods. According to your instructions, we revised the paper and took into account the changes and suggestions of the reviewers in our manuscript. You can find all changes and responses to each comment made by the reviewers in the following lines.
We would like to thank you and the reviewers for your very helpful comments.
Sincerely yours:
Esther Ramírez Moreno, PhD.
Reviewer 2
Open Review
(x) I would not like to sign my review report
( ) I would like to sign my review report
English language and style
(x) Extensive editing of English language and style required
( ) Moderate English changes required
( ) English language and style are fine/minor spell check required
( ) I don't feel qualified to judge about the English language and style
|
Yes |
Can be improved |
Must be improved |
Not applicable |
|
|
Does the introduction provide sufficient background and include all relevant references? |
( ) |
(x) |
( ) |
( ) |
|
Is the research design appropriate? |
( ) |
( ) |
(x) |
( ) |
|
Are the methods adequately described? |
( ) |
( ) |
(x) |
( ) |
|
Are the results clearly presented? |
( ) |
( ) |
(x) |
( ) |
|
Are the conclusions supported by the results? |
( ) |
( ) |
(x) |
( ) |
Comments and Suggestions for Authors
General comments for authors
The ms is improved but still does not have satisfied quality. Several suggestion can be given to provide novelty, for example to include investigations related to stability of powder during the storage, i.e. measurement of TPC; AO activity and color upon or during storage, as well as surveying of water activity changes.
The style of writing is inappropriate. Comments on Abstract are only an example that should give idea how to improve the whole text to be concise and adequate for publishing.
Abstract
Abstract of 350 words needs to be shortened for the sake of conciseness, exactness and clarity. It needs to be completely rewritten and checked by native speaker with experience in scientific writting.
Answer: The abstract was shortened and improved with the most important and precise information and checked by a native speaker.
Following sentences “Mexico is undergoing an epidemiological and nutritional transition that generates an increase in the national morbidity and mortality rate. This effect can be reduced by the promotion of a varied and inclusive diet including indigenous foods, such as underutilized xoconostle fruit, with a content of bioactive compounds, which contributes to prevention of different diseases.” (50 words) needs to be avoided since relation between national mortality and an attempt to dehydrate local fruit by time consuming and very costly freeze drying is not remarkable enough.
Answer: The sentence had been eliminated. We agree with the reviewer about the importance of consumption and the relation of this diseases is with the fresh fruit no with a dry sample.
There are many unnecessary parts in Abstract. Following information such as that fruits were provided by an association of farmers, that all the data were expressed as dry matter etc are uncommon to exist in Abstract and should be avoided as well.
Answer: This information was a recommendation of the reviewer 1 in the previous revision, however we agree that the abstract must be rewritten and consider only the information more important related with the results with this study.
The sentence “Xoconostle fruits were characterized by an attractive color between yellow-orange (cv Cuaresmeño) and reddish rose (cv. 33 Rosa)” does not provide info about the work done and should not exist.
Answer: The information had been eliminated.
Also widely accepted facts needs to be avoided such as “it is more important to take advantage of the component in the whole dehydrated product than when the extract of the fruits is used”. Also, this is not a topic and any such comparison was made during the study conducted.
Answer: The information had been eliminated.
The sentence “The powder fruits had properties for the formation of gels, retention of water, lipids and glucose” should be rewritten since fruits powder generally have such properties. Specific data obtained for analyzed fruits powders needs to be given instead.
Answer: The sentence had been completed with dates and the relevance with other studies.
The following sentence “Therefore, the samples could be considered sources of fiber and vitamin C” should be deleted since the last sentence in Abstract contains its repetition.
Answer: The sentence had been eliminated.
Also, the sentence “These samples also had a high content of phenolic compounds, which together with ascorbic acid, contributing to their antioxidant activity” should be rewritten. It is widely known that phenolics and ascorbic acid contribute AO activity, authors did not consider specific contribution of individual compounds to total AO activity in their work…..
Answer: The sentence had been rewritten.
The last sentence could be shorter. Expression such as great importance needs to be avoided.
Answer: The sentence had been eliminated.
More exact data should be introduced, for example duration of drying and temperature can be included in part of abstract related to drying (in brackets), dry matter of powder obtained (also in bracket), not with details, just number (For example Moisture of ….was achieved upon 96 h of freeze drying at temperature of ... ), data related to ratio of soluble and insoluble fibers, exact values of WHC i OHC, as well as FC, ABTS and DPPH. Some differences between cultivars might be mentioned, if significant enough.
Answer: We had included all the recommendations of the reviewer and the abstract had been changed.
MM
There are many confused parts of MM, with missing details or parts different than original version
For example, In original version vacuum was 0.040, in revised version 0.140?
Answer: We correct the mistake of the real vacuum used and specified the freeze-dryer equipment. The minimum set pressure range in these devices varies between 0.015 to 0.096 Mbar. Its depends of the used equipment.
Quantity (volume) of acid is missing in Five hundred mg of each sample were homogenized with metaphosphoric acid (4.5%), with magnetic stirring for 15 min in the dark and 126 centrifuged (9616 g, 15 min) (Thermo Scientific, Sorval RC-5, Waltham).
Answer: We included the information about the amount of acid used.
Since it would be extremely time consuming for reviewers to introduce all corrections necessary only some examples are given, there are numerous incorrectness.
Answer: The whole paper had been reviewed and corrected by the researchers and by a native speaker.
RD
For the sake of clarity many parts of RD should be changed as well.
Answer: The part of results and discussion had been reviewed and corrected.
In Table 1 the values for ash are impossible for the stats to be significantly different. Please check the stats again!
Answer: We check again all the statistics results and we corrected the differences. In effect, there are not differences between the cultivars in ashes.
Also, not all values should have 2 decimal places, but the last significant digit, it is impossible for all methods to have the same last significant digit, this applies to all tables.
Answer: We homogenized the form to present the results in the tables, only with a decimal after the point.
244-250, it is not clear what values correspond to which sample. The last sentence in this paragraph is completely unclear. Expressing results such as ≈ 30 g/100g dm is not acceptable. Exact numbers should be given, with SD included.
Answer: We included in the text the complete results and in the last sentence we complete the result of the study of Guzman et al. 2010, in order the sentence was clear:
Line 244-253 “Table 1 shows the values obtained for nutritional composition of the powdered whole xoconostle fruits (with peel and seeds) belonging to two cultivars analyzed in this study. The dried samples were characterized by a high proportion of total carbohydrates constituted by soluble sugars and dietary fiber. In addition, the values obtained for XC were lower in comparison with XR, regarding soluble sugars (9.8 ± 0.7 and 29.9 ± 0.5 g/100 g dm) and fiber (30.8 ± 0.7 and 36.8 ± 0.9 g/100 g dm). These samples had a lower proportion of fat (2.5 ± 0.1 and 11.9 ± 0.1 g/100 g dm), proteins (4.0 ± 0.0 and 4.8 ± 0.1 g/100 g dm) and ashes (11.3 ± 2.9 and 11.7 ± 1.5 g/100 g dm). These values of xoconostle powder had the same tendency that other study of the edible portion (peel and skin) of xoconostle fruit, where around 60 and 81 % were constituted of total carbohydrates and 26.7 to 29.5 % of total dietary fiber [12]”.
Line 275 unit is missing after 100
Answer: The unit missed was added.
Extensive checking of units through text is strongly recommended.
Answer: The units through the text was strongly reviewed.
Conclusion
Conclusion should be rewritten, the last sentence deleted.
Answer: The conclusion was rewritten and the last sentence was deleted